# Out-of-Pocket Health Expenditures Associated with Chronic Health Conditions and Disability in China

**DOI:** 10.3390/ijerph20156465

**Published:** 2023-07-27

**Authors:** Jingyi Gao, Hoolda Kim, Sophie Mitra

**Affiliations:** 1Department of Economics, Fordham University, Bronx, NY 10458, USA; 2College of Business and Economics, Fayetteville State University, Fayetteville, NC 28301, USA; hkim7@uncfsu.edu

**Keywords:** disability, chronic conditions, out-of-pocket expenditures, extra costs, health care, social protection

## Abstract

The objective of this study is to estimate the extra costs of living associated with chronic health conditions and disabilities in China. Leveraging the 2018 China Health and Retirement Longitudinal Study involving 13,530 respondents aged 50 and over, we apply both an ordinary least squares linear regression model and a logistic model to analyze the correlation between medical out-of-pocket expenditures (OOPEs) and chronic health conditions, as well as disabilities measured by Activities of Daily Living (ADL) and Instrumental Activities of Daily Living (IADL) limitations. This paper bridges the gap in the literature on OOPEs and their association with disabilities and chronic health conditions, respectively. We find that ADL limitations, IADL limitations, and chronic health conditions are consistently associated with higher OOPEs. The odds that older persons with disabilities and chronic health conditions incur OOPEs are two to three times higher than for persons without disabilities and chronic health conditions, respectively. Persons with disabilities and chronic health conditions have the highest OOPEs. The findings suggest that more policy and research attention is necessary to improve the financial protection of those with chronic health conditions and disabilities, including through access to comprehensive health insurance coverage.

## 1. Introduction

China has an aging population. Approximately 254 million people are aged 60 and over out of the total population of more than 1.4 billion people. This number is projected to grow to 402 million by 2040, which will account for 28% of the total population [1]. The prevalence rates of chronic diseases and functional limitations have been increasing along with the growing aging population [2]. In the global south, healthcare spending has been shown to be higher among households with disabilities than those with no disabilities using World Health Survey data [3]. People with disabilities are found to spend more on health care services, assistive devices, transportation, and daily activity assistance to achieve the equivalent living standard of people without disabilities [4]. Moreover, because it is difficult for the public health system to provide universal coverage in the global south [5], households with persons with disabilities may incur large out-of-pocket expenditures and become more vulnerable to catastrophic health expenditures [6], i.e., spend more than 40% of their income net of subsistence needs on health services. In China, catastrophic health expenditures are not uncommon, affecting 12.99% and 15.56 of middle-aged and older adults, respectively [7]. In addition, the substantial increase in healthcare costs in China has created challenges for low-income residents in recent years, with many unable to afford necessary healthcare services and with health expenditure-induced poverty [8]. It is important to provide an analysis of out-of-pocket health expenditures associated with disability and chronic health conditions among older adults in China to inform policies related to health and economic security.

There is extensive international literature examining the association between out-of-pocket expenditures (OOPEs) and health conditions. However, very little research has been conducted on the association between OOPEs and disabilities. While persons with disabilities are expected to spend more on health care services than their nondisabled counterparts, they may not incur high OOPEs due to low income or, more broadly, a lack of access to health care services.

This paper fills the gap in the literature by examining the association between OOPEs and disabilities and chronic health conditions among middle age and older adults in China. The aim of this article is to find out whether older Chinese with disabilities and chronic health conditions have more OOPEs than persons without using the nationally representative 2018 China Health and Retirement Longitudinal Study (CHARLS) dataset. We use two measures of functional disability that are commonly used among older adults, activities of daily living (ADL) and instrumental activities of daily living (IADL) limitations [9].

The ordinary least squares (OLS) linear regression and logistic regression models are employed to estimate the association between OOPEs on the one hand, and in turn, ADL limitations, IADL limitations, and chronic conditions, on the other. The study finds a positive association between the presence of ADL limitation/IADL limitation/multimorbidity and the odds of incurring OOPEs among middle-aged and older adults in China. The odds of incurring OOPEs for respondents with multimorbidity is around three times that of respondents without multimorbidity. ADL limitations, IADL limitations, and chronic conditions are significantly associated with higher OOPEs. A ten-unit increase in the chronic condition, ADL, and IADL indices is associated with 65%, 29%, and 25% increases in OOPEs, respectively.

The remainder of the paper is organized as follows. Section 2 reviews the literature on the extra costs of living with a disability. Section 3 describes the dataset, measures, and methods. Section 4 presents the results. Section 5 discusses the results and concludes.

## 2. Literature Review

An extensive international literature base has been developed to estimate the extra costs of living with a disability. Several studies have used a standard of living approach, which is an indirect method designed to identify the changes in the relationship between income and utility to estimate the extra costs of living with a disability [10,11,12,13]. Schuelke et al. (2021) compare the cost of living for households with disabled members and without disabled members in the UK and found that the average extra cost of living with a disability accounts for half of the median household income per week from 2013 to 2015 [10]. Similarly, for the United States, Morris et al. (2021) found that disabled households require 28% more income to obtain the same standard of living as a comparable household without a disabled member [11]. Roddy (2021) used the standard of living approach to investigate the extra cost of living with a disability for families with a child with a chronic illness/disability in Ireland. The results indicate that families with children with chronic illnesses or disabilities are more likely to experience financial hardship and considerable disadvantages across all levels of disability and income [12]. Amin and Adros (2019) measure the magnitude of the extra cost of disability in Malaysia. They found that the extra costs of disability for households living with at least one person with disabilities account for 27.5% of their monthly income [13]. Palmer et al. (2019) used the standard of living approach and found that the direct cost of disability for households in Cambodia is 19% of their monthly consumption expenditure [14]. Mitra et al. (2017), after conducting a systematic literature review on the extra costs of disability in ten countries, suggest that the extra costs of disability are the highest among persons with severe disabilities who are living alone or in a small-sized household [15].

Limited research has been conducted on this topic in China. Loyalka et al. (2014) measured the extra costs of disability across different types of disability and different types of households in both urban and rural areas of China and found a strong negative association between disability and household income [16]. Despite many studies that estimate the extra costs of living with a disability, in general, limited attention has been drawn to OOPE, which is a specific source of extra costs associated with healthcare services.

There is a small but growing literature on OOPEs and disability. Hanass-Hancock and Deghaye (2016) document OOPEs for adults with different types and degrees of disability covering categories for assistive devices, care and support for daily living, accommodation and travel to work, healthcare services, and general travel and additional cost in South Africa [17]. In addition, Lee et al. (2016) found that the OOPEs for households with disabled members are 1.29 times higher than for households without disabled members in Korea using the 2010–2011 Korean Health Panel dataset. Several studies have applied two-part regression models to assess the relationship between functional limitations and OOPEs [18]. For instance, Nguyen et al. (2021) found that the severity level of disability is positively associated with OOPEs. Adults with disability are more likely to experience a high burden of OOPEs [19]. Salinas-Rodriguez et al. (2020) found that the presence of ADL and IADL limitations is positively associated with total annualized OOPEs among older Mexican adults aged 60 and older [20].

In contrast, the literature on the relationship between OOPEs and chronic conditions is large. Lee (2014) found evidence of higher OOPEs among non-institutionalized adults with chronic conditions in the United States using the 2010 Medical Expenditure Panel Survey. However, this study provides only descriptive statistics. It does not use a multivariable model [21]. Callander et al. (2017) used linear and logistic regression models and found a positive association between chronic conditions and household OOPEs among adults in Australia [22]. Three studies investigate the association between chronic conditions and OOPEs in China. Zhao et al. (2021) used a quantile regression analysis and the 2015 CHARLS data to investigate the relationship between chronic diseases and OOPEs across each quantile. They found that multimorbidity is positively associated with OOPEs and is more substantive at the upper tail of the health expenditure distribution [23]. Zhao et al. (2021) used the 2011 and 2015 CHARLS data to compare the OOPEs of a single physical condition with mental–physical multimorbidity. They found a positive association between mental–physical multimorbidity and OOPEs in China [24]. Lan et al. (2018) applied logistic regression to assess the association between chronic conditions and out-of-pocket health payment-induced poverty and claimed that out-of-pocket health payment-induced poverty increases rapidly with the growing number of households suffering from chronic diseases in Shaanxi Province, China [25].

This paper is at the intersection of two literature groups: the literature on OOPEs and the literature on the extra costs of living with a disability. The literature on OOPEs has mostly focused on the association between OOPEs and chronic conditions. It has not been linked to the literature on the extra costs of living with a disability. This study aims to examine whether and to what extent chronic conditions and disabilities are associated with OOPEs with a focus on middle-aged and older Chinese individuals.

## 3. Data and Methods

This paper uses the 2018 China Health and Retirement Longitudinal Study (CHARLS), which is a nationally representative longitudinal survey of people aged 45 and over and their partners in China. The CHARLS datasets provide demographic, socioeconomic, and health information at the individual and household levels. The unit of analysis is the individual and individual cross-sectional weights with household and individual non-response adjustment applied to all results.

As ADL and IADL questions were not asked among respondents aged 45 to 49, we focus on individuals aged 50 and above. Out of a total of 17,255 respondents aged 50 and above, 13,530 individuals had no missing values on OOPEs, ADL, IADL, chronic health conditions, and control variables and are the focus of this study. For chronic health conditions, where data was missing for more than 5% of observations, we analyzed whether data were missing randomly. In particular, we ran a logistic regression model where the dependent variable was equal to one for data missing on chronic health conditions and zero otherwise. Independent variables included OOPEs, as well as the control variables of the main analysis (see Equations (1) and (2) below). Being 70 and older (AOR = 2.30, 95%CI = 1.98, 2.68) and not being married (AOR = 1.56, 95%CI = 1.36, 1.79) were associated with higher odds of having missing data on chronic health conditions. Having disproportionately more missing data for demographic subgroups is a major limitation of the study.

### 3.1. Out-of-Pocket Expenditures (OOPEs)

OOPEs are defined as the sum of total expenditures for outpatient and inpatient care after deducting any amount covered by health insurance. CHARLS measures individual outpatient and inpatient out-of-pocket expenditures in the past year.

CHARLS is a sister study of the Health and Retirement Study (HRS). When collecting data on OOPEs, CHARLS adopted a questionnaire structure and approach that closely align with HRS. These questions follow a standardized pattern that has been widely utilized in surveys and documented in the literature on OOPEs [26]. All respondents are asked to report the number of inpatient visits and the total expenditure of inpatient care in the previous year [8].

Respondents are also asked to report the number of outpatient visits in the last month and the amount of out-of-pocket expenditures that they paid for doctor’s visits, including the fees paid for treatment, medication costs, and prescription drugs. As only last month’s out-of-pocket expenditures for doctor’s visits are measured, we multiply its value by 12 to approximate the annual outpatient expenditure. This has been conducted in the literature on OOPEs, in general, and using CHARLS, in particular [24].

We use two variables for OOPEs. The first one is a binary dependent variable, which takes a value of one if the OOPES for an individual are greater than zero and zero otherwise. The second one takes the natural logarithm of OOPEs.

### 3.2. Activities of Daily Living (ADL) and Instrumental Activities of Daily Living (IADL)

Disability was measured using difficulties in activities of daily living questions. Such questions were initially developed to capture the physical effects of aging, and, as such, in some surveys, the questions are only administered to respondents above a certain age—50 and above in CHARLS. Since the 1980s, they have been used in a variety of clinical, policy, and research contexts [27,28,29,30]. There are different types of activities of daily living questions, including basic and instrumental. Basic ones are fundamental for body functioning (e.g., walking a specific distance) and include self-care activities, such as feeding oneself, while instrumental ones cover more complex tasks, such as managing money.

CHARLS dataset contains six questions that are designed to assess the limitations in basic activities of daily living (ADLs), including bathing, dressing, eating, getting in/out of bed, using the toilet, and controlling urination. Five questions are designed to assess limitations in Instrumental Activities of Daily Living (IADLs), including using the phone, managing money, taking medications, shopping for groceries, and preparing hot meals. We use a binary variable for ADL/IADL that takes a value of one if an individual has one or more ADL or IADL limitations and zero otherwise. In addition, the ADL and IADL indices are constructed by normalizing the number of ADL and IADL limitations to 100 for each individual as follows.

To capture the number and severity of ADL or IADL limitations, we use ADL and IADL indices as developed by Stewart and Ware (1992; page 80) [29] and used in the literature since [31]. The ADL index (from 0 to 100) is the normalized index of the sum of the answers (each equal to 1 (No, I do not have any difficulty), 2 (I have difficulty but can still do it), 3 (Yes, I have difficulty and need help) or 4 (I cannot do it)) to the six ADL questions as follows with a minimum of six and a maximum of 24: ADL Index = (SumADL − 6)/(24 − 6) × 100. For instance, if someone answers 1 (No, I do not have any difficulty) to the six ADL questions, the sum of answers (SumADL) is six, and the ADL index is zero. If someone answers 1 (No, I do not have any difficulty) to five ADL questions but a 4 (I cannot do it) to one ADL question, the sum of answers (SumADL) is 9, and the score is (9 − 6)/(24 − 6) × 100 = 1/6 × 100.

Similarly, the IADL index is the normalized score or sum of the answers (each equal to 1, 2, 3, or 4) to the five IADL questions as follows with a minimum of five and a maximum of 20: IADL Index = (SumIADL − 5)/(20 − 5) × 100.

### 3.3. Multimorbidity

Questions on chronic conditions are diagnoses by medical professionals, as reported by the respondents. Respondents are asked whether they have ever been told by a doctor that they have any of the 15 chronic conditions, including hypertension, diabetes, cancer, chronic lung disease, heart disease, stroke, psychiatric problem, arthritis, dyslipidemia, liver disease, kidney disease, digestive disease, asthma, depression, and memory problem. The multimorbidity binary variable takes a value of one if an individual has ever been diagnosed with two or more chronic diseases and zero otherwise, as defined in the multimorbidity literature [24].

To capture the number of chronic health conditions, a non-communicable disease (NCD) index is constructed by normalizing the sum of chronic conditions to 100 for each respondent in a way similar to the ADL and IADL indices above: NCD = (SumChronicConditions)/(15) × 100.

### 3.4. Covariates

We include age, gender, marital status (married and partnered, unmarried, and others), education (illiterate, primary school, secondary school, college and above), and Hukou-residence combination (rural Hukou/rural residence, rural Hukou/urban residence, urban Hukou/urban residence, and urban Hukou/rural residence) as control variables for the regression analysis. Hukou is the Chinese residency registration system. People are legally required to register as residents of a particular area, and it is categorized as rural Hukou or urban Hukou.

We include this set of controls as these are relevant determinants of economic outcomes, in general, and OOPEs, in particular. Gender, age, and marital status are demographic variables that are typically included in research on the extra costs of disability [15]. For China, Hukou and residence are standard controls in studies of economic outcomes [23,24]. The China household registration system is called “Hukou”. Individuals born in rural areas receive “rural hukou”, while those born in cities have “urban hukou”. The migration of workers from rural to urban areas started in the early 1980s as a response to labor market deregulation and demand for services in cities. So many individuals reside in urban areas while having a rural Hukou.

Hukou, residence, as well as age, are particularly relevant as they affect health insurance coverage. China’s basic national health insurance system is related to hukou. In 2003, the New Cooperative Medical System (NCMS) was launched by the Chinese government, which aimed to open basic health services to older people in order to relieve the burden on rural residents. The NCMS covered rural residents while employed urban residents were insured by the Urban Employee Basic Medical Insurance (UEBMI), and urban residents without formal employment and students were insured by the Urban Resident Basic Medical Insurance (URBMI) [32].

### 3.5. Models

We employ two models to examine the association between OOPEs and health problems: a logistic regression and an OLS linear regression, as per Equations (1) and (2) below.
(1)$$D_{i}=a_{0}+a_{1}H_{i}+\sum_{j=2}^{6}a_{j}X_{ji}+{\;\varepsilon }_{i}$$
(2)$$\;Y_{i}={\;b}_{0}+b_{1}H_{i}+\sum_{j=2}^{6}b_{j}X_{ji}+{\varepsilon^{\prime }}_{i}$$
where, in both equations, *i* indexes individuals, and $X_{ji\;}$ is a vector of five control variables at the individual level as follows: education, gender, marital status, age, hukou/residence. In Equation (1), $D_{i}$ is the binary OOPEs indicator and $H_{i}$ is a binary indicator measure of disability or health through ADL, IADL, or multimorbidity binaries. In Equation (2), $Y_{i}$ is the natural logarithm of OOPEs in US dollars and $H_{i\;}$ is a measure of disability or health through an ADL Index, IADL Index, or NCD Index.

In Equations (1) and (2), respectively, ε_*i*_ and ε′_*i*_ represent the residual or error term, which accounts for the unexplained variation in the dependent variable that is not captured by the independent variables.

The OLS linear regression model (1) assumes a linear relationship between the dependent variable, log of OOPEs, and independent variables, including the ADL, IADL, and NCD indices. In contrast, the logistic regression model does not assume a linear relationship between the dependent variable, incurring OOPEs, and the independent variables, including ADL, IADL, and multimorbidity status. Both models require no multicollinearity between controls and use the same set of control variables. Stata/BE 18.0 is used in this study to run the regressions.

## 4. Results

### 4.1. Descriptive Statistics

Table 1 presents the share of individuals with strictly positive OOPEs and average OOPEs based on individuals’ ADL limitation, IADL limitation, and multimorbidity status. On average, individuals incur 547.40 USD per year for outpatient and inpatient care, and 27% of individuals have strictly positive OOPEs. More respondents with ADL limitation, IADL limitation, or multimorbidity have strictly positive OOPEs compared to those without, and their average OOPEs are higher. For instance, individuals with IADL limitations spend, on average, 1144.4 USD on OOPEs compared to 374.70 USD for those without IADL limitations.

There may be overlap between the groups of individuals with ADL limitation, IADL limitation, or multimorbidity. Figure 1 shows mean OOPEs for those with different combinations of ADL limitations, IADL limitations, and multimorbidity. The group of respondents who experience ADL and IADL limitations as well as multimorbidity had the highest mean OOPEs (1501.93 USD per year), implying that the ADL and IADL limitations combined with multimorbidity may induce high OOPEs. Figure 2 illustrates the share of individuals who used health services and incurred strictly positive OOPEs in the past year based on their ADL/IADL limitations and multimorbidity status. Individuals with multimorbidity utilize healthcare services and incur OOPS more than those with ADL/IADL limitations. For instance, 29.8% of individuals with multimorbidity and no ADL/IADL limitations incur OOPS compared to 24.4% and 25.3% of those with ADL and IADL, respectively (and no multimorbidity).

Table 2 displays the sociodemographic characteristics of respondents, which include gender, marital status, education, and Hukou/residence status. Female participants represent 50.6% of all respondents. Regarding marital status, 84.5% of the respondents are married or partnered. More than half of the respondents reside in rural areas and hold rural Hukou. Table 2 also reports the sociodemographic characteristics of the respondents with ADL limitation, IADL limitation, and multimorbidity. The demographic composition is calculated for each group of people with the presence of ADL limitation, IADL limitation, and multimorbidity. People with ADL limitation, IADL limitation, and multimorbidity are more likely to be female and illiterate. People with ADL or IADL limitations are more likely to have rural Hukou and residence.

### 4.2. Model Results

Table 3 presents the results of the logistic regression analysis. The presence of ADL limitation, IADL limitation, and multimorbidity is found to be positively associated with incurring annual OOPEs after controlling for age, gender, marital status, education, and Hukou/residence of living. The odds of incurring OOPEs for respondents with ADL limitations was 2.22 times that of respondents without ADL limitations, indicating a higher chance of incurring OOPEs for respondents with ADL limitations. The odds of incurring OOPEs for respondents with IADL limitations was 2.21 times that of respondents without IADL limitations. The odds of incurring OOPE for respondents with multimorbidity was 3.17 times that of respondents without multimorbidity.

The results of the linear regression analysis are in Table 4. Similar to the results in Table 3, we find a positive relationship between OOPEs and the ADL, IADL, and NCD indices after controlling for age, gender, marital status, education, and Hukou/residence of living. The estimates indicate that a ten-unit increase in the ADL index is associated with a 29% increase in OOPEs. A ten-unit increase in the IADL index is associated with a 25% increase in OOPEs. A ten-unit increase in the chronic condition index is associated with a 65% increase in OOPEs.

## 5. Discussion

The study examines the association between OOPEs and the presence of ADL, IADL, and multimorbidity among middle-aged and older adults in China. We found that the presence of ADL limitations, IADL limitations, or multimorbidity is associated with higher odds of incurring OOPEs and with higher OOPEs. Descriptive statistics suggest that people with ADL limitations, IADL limitations, and multimorbidity are the group that, on average, spends the most on OOPEs. Results from multivariate regressions suggest that those with ADL limitations, IADL limitations, or multimorbidity have two to three times the odds of incurring OOPEs after controlling for age, sex, marital status, education, and Hukou/residence.

The odds of incurring OOPEs for people with multimorbidity were found to be more than three times higher than for those without multimorbidity. An increase in the chronic condition index was significantly associated with a greater amount of OOPEs from the regression results. These results are similar to the findings of Zhao et al. (2021), that found a statistically significant positive association between the number of chronic conditions and OOPEs using the 2015 CHARLS data [24].

We also found that the odds of incurring OOPEs for people with disabilities was more than twice that of people without disabilities as well as a positive correlation between the ADL/IADL indices and OOPEs. This result is consistent with the findings for Vietnam by Nguyen et al. (2021) that suggest a positive association between the severity level of disability and OOPEs: adults with mild, moderate, and severe disabilities have additional out-of-pocket expenses of 108,879, 181,682, and 353,283 VND per month, respectively, compared to those without a disability [19].

The results of this study have several limitations. First, this paper focuses on the association between ADL/IADL/chronic conditions and OOPEs. It does not provide a precise assessment of the causal effect of ADL/IADL/multimorbidity on OOPEs. It also cannot address the dynamic linkages between chronic conditions and disability, as the former may lead to the latter, which may have implications for healthcare utilization and OOPEs. Persons with chronic conditions but limited access to healthcare may end up with preventable ADL or IADL limitations. Further research is needed to examine the dynamic relationship between chronic health conditions, disabilities, and OOPEs by exploiting the CHARLS panel data. In addition, this study is restricted to a CHARLS sample for one year collected in 2018. While we also conducted the analysis with the 2015 CHARLS data, and the results were similar, due to the unavailability of data after 2018, we cannot test how the association between OOPEs and ADL/IADL/multimorbidity might have changed during the COVID-19 pandemic.

## 6. Conclusions

This paper makes several contributions. To our knowledge, it is the first study to examine the association between ADL/IADL limitations and OOPEs in China. The paper contributes to the literature on the extra costs of living with a disability [15,16] and provides new estimates for China on health-related extra costs. Second, this paper bridges the literature on the extra costs of living with a disability [15,16] and the literature on OOPEs and chronic conditions [17,18,19,20,21,22,23,24,25] by comparing OOPEs of individuals who have ADL limitations, IADL limitations, and multimorbidity.

While the odds of incurring OOPEs were found to be relatively lower based on ADL and IADL status than based on multimorbidity status, they were significantly higher for all three groups than among persons with no disabilities or health conditions. We also found that people with ADL and IADL limitations, as well as multimorbidity, have the highest OOPEs on average. This result warrants further research, in general, and in China, in particular, and suggests that the literature on the extra costs of living with a disability should consider multimorbidity as a risk factor for higher extra costs. This paper finds evidence of the financial burden of living with chronic health conditions and disabilities among middle-aged and older Chinese people. The findings suggest that more policy and research attention is necessary to improve the financial protection of those with chronic health conditions and disabilities, including through access to comprehensive health insurance coverage.

## Figures and Tables

**Figure 1 ijerph-20-06465-f001:**
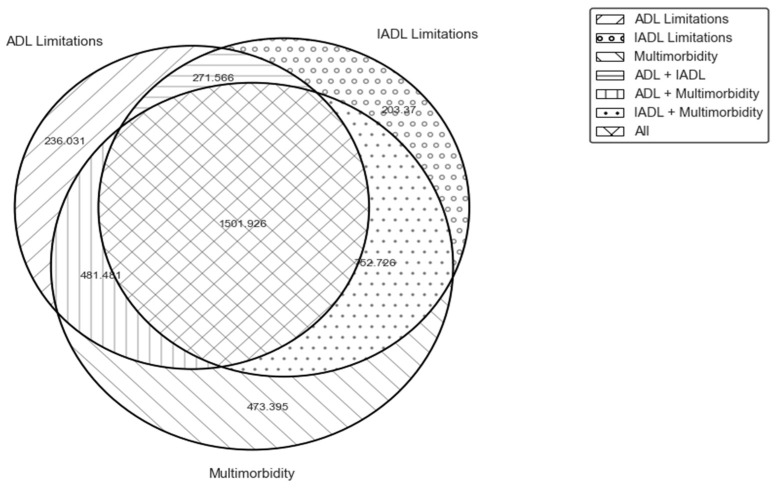
Mean OOPEs in dollars in the past year by ADL limitation/IADL limitation/Multimorbidity status. Source: Authors’ calculations based on 2018 CHARLS data.

**Figure 2 ijerph-20-06465-f002:**
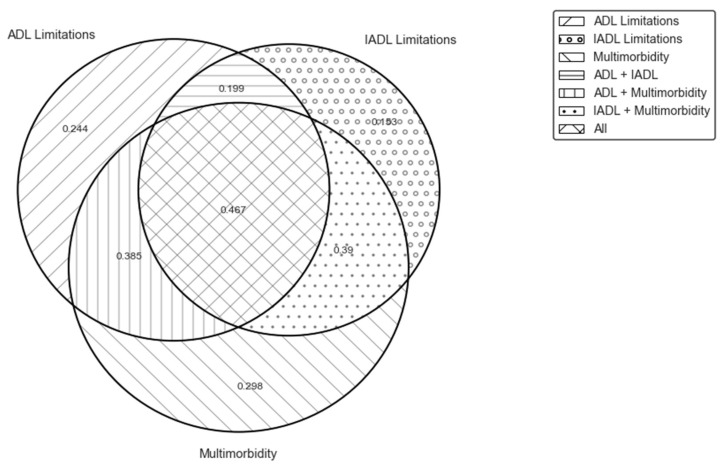
Percentage of individuals with strictly positive OOPEs in the past year by ADL/IADL/multimorbidity status. Source: Authors’ calculations based on 2018 CHARLS data.

**Table 1 ijerph-20-06465-t001:** Mean out-of-pocket health expenditures by ADL limitation/IADL limitation/Multimorbidity status.

	All	ADL Limitation			IADL Limitation			Multimorbidity		
		Yes	No	Diff.	Yes	No	Diff.	Yes	No	Diff.
n	13,530	2607	10,923		3237	10,293		8810	4720	
Mean of OOPEs (US$)	547.50 (43.00)	1155.54 (186.59)	414.29 (32.66)	741.25 *** [0.00]	1144.39 (156.94)	374.65 (31.42)	769.74 *** [0.00]	720.52 (60.80)	223.54 (48.00)	496.98 *** [0.00]
% Incurring OOPEs	0.27 (0.00)	0.41 (0.01)	0.24 (0.01)	0.17	0.40 (0.01)	0.23 (0.01)	0.17	0.34 (0.01)	0.14 (0.01)	0.20

Notes: Standard errors in parentheses and *p*-values in brackets; *** *p* < 0.01. Estimates are weighted using individual weights with household and individual non-response adjustment. Source: Authors’ calculations based on 2018 CHARLS data.

**Table 2 ijerph-20-06465-t002:** Descriptive statistics by ADL limitation/IADL limitation/Multimorbidity status.

	All	ADL Limitation	IADL Limitation	Multimorbidity
Age (years)				
50–59	0.368	0.199	0.220	0.318
60–69	0.385	0.391	0.380	0.402
70 and above	0.246	0.410	0.400	0.280
Gender				
Male	0.494	0.402	0.383	0.470
Female	0.506	0.598	0.617	0.530
Marital status				
Married or partnered	0.845	0.758	0.758	0.831
Unmarried and others	0.155	0.242	0.242	0.169
Education				
Illiterate	0.385	0.530	0.557	0.400
Primary school	0.231	0.237	0.219	0.230
Secondary school	0.327	0.205	0.195	0.313
College and above	0.057	0.028	0.030	0.057
Hukou/Residence				
Rural Hukou/Rural Residence	0.514	0.612	0.608	0.506
Rural Hukou/Urban Residence	0.216	0.188	0.187	0.211
Urban Hukou/Urban Residence	0.245	0.173	0.185	0.257
Urban Hukou/Rural Residence	0.026	0.027	0.021	0.026
Observations	13,530	2607	3237	8810

Note: Estimates are weighted using individual weights with household and individual non-response adjustment. Source: Authors’ calculations based on 2018 CHARLS data.

**Table 3 ijerph-20-06465-t003:** Logistic regression of incurring OOPEs (odds ratios).

	(1)	(2)	(3)
	Binary OOPEs	Binary OOPEs	Binary OOPEs
ADL indicator	2.220 ***		
	(0.129)		
IADL indicator		2.208 ***	
		(0.119)	
Multimorbidity indicator			3.166 ***
			(0.199)
Age (years) (Ref: 50–59)			
60–69	0.952	0.960	0.900 *
	(0.057)	(0.057)	(0.056)
70 and above	1.200 **	1.190 **	1.176 **
	(0.084)	(0.083)	(0.084)
Gender (female)	1.153 ***	1.136 **	1.132 **
	(0.063)	(0.062)	(0.063)
Marital status (unmarried and others)	0.940	0.933	0.965
	(0.065)	(0.065)	(0.063)
Education status (Ref: Illiterate)			
Primary school	1.144	1.188 ***	1.200 *
	(0.073)	(0.075)	(0.074)
Secondary	1.147 **	1.198 **	1.094
	(0.080)	(0.084)	(0.079)
College and above	1.238	1.290 *	1.159
	(0.169)	(0.175)	(0.159)
Hukou/Residence (Ref: Rural Hukou/Rural Residence)			
Rural Hukou/Urban Residence	0.996	1.002	0.953
	(0.069)	(0.069)	(0.069)
Urban Hukou/Rural Residence	1.312 **	1.358 **	1.288 **
	(0.162)	(0.166)	(0.163)
Urban Hukou/Urban Residence	1.238 ***	1.227 ***	1.127
	(0.091)	(0.090)	(0.086)
Intercept	0.248 ***	0.234 ***	0.140 ***
	(0.017)	(0.017)	(0.012)

Notes: Robust standard errors in parentheses, *** *p* < 0.01, ** *p* < 0.05, * *p* < 0.1. Estimates were weighted using individual weights with household and individual non-response adjustment. Source: Authors’ calculations based on 2018 CHARLS data.

**Table 4 ijerph-20-06465-t004:** Linear regression of the natural logarithm of OOPEs.

	(1)	(2)	(3)
	OOPEs	OOPEs	OOPEs
ADL index	0.029 ***		
	(0.002)		
IADL index		0.025 ***	
		(0.002)	
NCD index			0.065 ***
			(0.002)
Age (years) (Ref: 50–59)			
60–69	−0.035	−0.041	−0.190 ***
	(0.073)	(0.073)	(0.072)
70 and above	0.225 **	0.168 *	0.067
	(0.091)	(0.091)	(0.089)
Gender (female)	0.175 **	0.148 **	0.087
	(0.069)	(0.068)	(0.067)
Marital status (unmarried and others)	−0.111	−0.081	−0.093
	(0.091)	(0.091)	(0.090)
Education status (Ref: Illiterate)			
Primary school	0.238 ***	0.269 ***	0.181 **
	(0.081)	(0.080)	(0.081)
Secondary	0.194 **	0.237 ***	0.159 *
	(0.086)	(0.086)	(0.085)
College and above	0.279	0.336 *	0.238
	(0.190)	(0.188)	(0.180)
Hukou/Residence (Ref: Rural Hukou/Rural Residence)			
Rural Hukou/Urban Residence	−0.027	−0.024	−0.043
	(0.081)	(0.080)	(0.081)
Urban Hukou/Rural Residence	0.368 **	0.405 **	0.229
	(0.169)	(0.169)	(0.168)
Urban Hukou/Urban Residence	0.368 ***	0.352 ***	0.193 *
	(0.101)	(0.099)	(0.099)
Intercept	1.206 ***	1.154 ***	0.484 ***
	(0.085)	(0.085)	(0.089)

Notes: Robust standard errors in parentheses, *** *p* < 0.01, ** *p* < 0.05, * *p* < 0.1. Estimates are weighted using individual weights with household and individual non-response adjustment; Source: Authors’ calculations based on 2018 CHARLS data.

## Data Availability

Publicly available datasets were analyzed in this study. This data can be found here: (https://charls.charlsdata.com/pages/Data/2015-charls-wave4/zh-cn.html, accessed on 4 July 2023).

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
