# Peer review of "Out-of-Pocket Health Expenditures Associated with Chronic Health Conditions and Disability in China"

_ijerph, 2023, doi:10.3390/ijerph20156465_

Round 1

Reviewer 1 Report

In the Introduction, it is necessary for the authors to better justify the conduct of the study.

The article should provide more detailed information about the objectives or research questions that this study aims to address using the 2015 China Health and Retirement Longitudinal Study (CHARLS).

It is important to clarify the methodology used to select the sample of 9,592 individuals aged 50 and over in 2015, specifically how individuals with missing values were handled and excluded from the analysis.

It is necessary to describe the process of applying individual cross-sectional weights to the results and explain how these weights were derived.

The article should provide more information about the questions related to out-of-pocket expenditures (OOPEs) in the CHARLS survey, including their reliability and validity, and whether they were adapted from existing measures or developed specifically for CHARLS.

The authors should clarify how health insurance coverage was assessed and accounted for when calculating OOPEs. Specifically, explain how the amount covered by health insurance was determined and deducted from total expenditures.

The methodology used to approximate annual outpatient expenditure by multiplying the last month's expenditure by 12 should be further justified and supported by previous research or validation studies.

It is essential to provide information on the validity and reliability of the questions used to assess activities of daily living (ADL) and instrumental activities of daily living (IADL) in the CHARLS dataset. Additionally, the article should indicate whether these questions have been used in previous studies and if they are widely accepted measures.

The normalization process used to construct the ADL and IADL indices should be described in detail. Explain how the number of limitations was transformed to a scale of 0 to 100 and provide a reference for this normalization approach.

The article should specify whether the questions regarding chronic conditions in the CHARLS survey were self-reported or diagnosed by medical professionals. If self-reported, discuss the potential limitations and biases associated with self-reported data.

It is important to mention any evidence or validation studies supporting the construct validity of the non-communicable disease (NCD) index constructed from the number of diagnosed chronic conditions.

The article should provide a clear definition and operationalization of the binary variable for multimorbidity, including the rationale for choosing two or more chronic diseases as the threshold.

The rationale for selecting the covariates, including age, gender, marital status, education, and Hukou-residence combination, should be provided. Discuss how these variables are relevant to the research objectives and previous literature supporting their inclusion.

Provide more details about the logit regression and OLS linear regression models, including the specific equations used, assumptions made, and the statistical software employed.

The discussion needs to be redone and does not follow an academic narrative by not contrasting with other studies.

Likewise, the authors should review the limitations described and include some others that have not been considered, such as recall or courtesy bias.

Extensive editing of English language required.

Reviewer 2 Report

Thank you for the opportunity to review this manuscript. This interesting study analyses the association between chronic conditions and out-of-pocket expenditures, Activities of Daily Living and Instrumental Activities of Daily Living in China. Comments for improvement are below:

Introduction:

1. Introduce the concept of ADL and IADL: What do these indicators measure, and their differences? Also, did CHARLS adopt any validated tool to collect information on these indicators?  

Methods

2. Explain the exclusion of records with missing values and whether any statistical approach would be applied to address this issue.

3. Include information on the currency conversion and updated costs (e.g., inflation adjustments).

Results

4. Explain the figures on healthcare utilization (consistently below one). Is it the number of visits and inpatient care during the last year? 

Discussion

5. The study found that OOPEs are higher among people with multimorbidity compared with those with ADL or IADL limitations. However, multimorbidity also causes ADL and IADL limitations. So, this comparison does not make sense to me. My suggestion is to include some reflections on this in this section.

6. In the discussion section, it would be valuable to include some reflections on using a 6-year-old dataset (2015-1017) and how these findings may or may not reflect socio-economic changes in China during the last years, mainly due to the COVID-19 pandemic.   

Minor comments

- Some references are missed: 40% threshold of catastrophic health expenditures (line 40); ADL and IADL approaches (line 53 and section 3.2)

- Lines 56-65 in the introduction section are part of the results section

- The colours of figures 1 and 2 do not match the colours of the legends.

I suggest some edits, but the quality is good overall.

Round 2

Reviewer 1 Report

Thank you for correcting the observations.

Moderate editing of English language required